# Preparation of Resistant Starch Types III + V with Moderate Amylopullulanase and Its Effects on Bread Properties

**DOI:** 10.3390/foods13081251

**Published:** 2024-04-19

**Authors:** Jianglong Li, Faxiang Deng, Peng Han, Yangyue Ding, Jianxin Cao

**Affiliations:** Faculty of Food Science and Engineering, Kunming University of Science and Technology, Kunming 650500, China; 113832401pp@163.com (J.L.); dengfaxiang0123@163.com (F.D.); dingyangyue77@163.com (Y.D.); jxcao321@hotmail.com (J.C.)

**Keywords:** bread, dietary food, mineral bioavailability, moderate pullulanase, pea resistant starch

## Abstract

The potential of PulY103A (a moderate amylopullulanase originating from *Bacillus megaterium*) for resistant starch production under moderate conditions (40 °C; a pH of 6.5) was investigated. PulY103A was much more suitable for pea resistant starch production with a high growth rate of 3.63. The pea resistant starch (PSpa) produced with PulY103A had lower levels of swelling power and solubility and a better level of thermostability than native pea starch (PSn) and autoclaved PS (PSa). The starch crystallinity pattern was B + V, which indicated that the PSpa belonged to RS types III + V. In addition, PSpa was used for breadmaking. The results showed that the bread quality was not significantly influenced compared to the control group when the content of PSpa was under 10% (*p* > 0.05). The bread supplemented with 10% PSpa had a significantly increased TDF content compared to that of the control (*p* < 0.05). Moreover, the in vitro mineral bioavailability of the bread sample was influenced gently compared to other dietary fibers, and the bread sample changed from a high-glycemic-index (GI) food to a medium-GI food corresponding to white bread at the same concentration of PSpa. These results indicated that PSpa is a good candidate for the production of dietary foods.

## 1. Introduction

Resistant starch (RS) is a kind of starch that cannot be digested by enzymes in the small intestine. RS is classified into five groups, including RS types I–V, i.e., RS type I, physically inaccessible starch; RS type II, uncooked or partially cooked native starch from green banana and sweet potato; RS type III, retrograded or recrystallized starch; RS type IV, chemically modified starch; and RS type V, amylose–lipid complex starch [1]. Due to its environmental friendliness and cost effectiveness, RS production by the enzymatic method is preferred, especially with moderate enzymes. Pullulanase is the main enzyme used in RS production. However, there have been few reports on moderate pullulanases applied for RS production [2,3,4].

RS plays an important role as dietary fiber in people’s diets and in the texture of functional food, which is different from traditional ones [5]. Five kinds of RS have been applied in dietary foods [6,7,8]. Among these dietary foods, bread is consumed in many different forms around the world as a source of starch, minerals, and fiber. For better textural and nutritional properties, bread has been made from refined wheat flour by mixing it with gluten, other cereals, and different types of RS. Nowadays, four types of RS (viz. RS type I, RS type II, RS type III, and RS type IV) are used in the baking of bread [7,9]. However, it was found that the sensory and textural properties of bread are negatively affected by increasing the RS content, such as through crumb stiffening and darker colors [10]. Also, some research has found that several RSs at certain levels do not significantly affect the sensory quality of bread [9].

Legumes are one of the resources used for starch production. Among legume starches, pea starch is a good material for RS production, and pea RS has better properties compared to native starch [3]. However, there have not been any reports on pea RS in breadmaking. There are several reports on pea RS prepared by the enzymatic method [11], but there is only one report about a pea resistant starch being prepared by a moderate enzyme (PulY103B, type I pullulanase, from *Bacillus megaterium*) [3]. That study showed that the pea RS produced by PulY103B was efficient with a high growth rate, and the RS had a special structure and properties, including low levels of swelling power and solubility and good thermostability [3]. In addition, our research group found a moderate amylopullulanase (PulY103A) originating from *Bacillus megaterium* with α-1,4-transfer activity. The enzyme has optimal reaction conditions at a pH of 6.5 and 45 °C, with almost 70% of its maximum activity being retained at 35 °C, indicating its potential in RS production under moderate conditions [12]. In this study, the capacity of pea RS production by PulY103A was investigated. In addition, the properties of RS and its potential applications in bread were also studied.

## 2. Materials and Methods

### 2.1. Materials

Pea starch (PS) was provided by Liangrun Food Co., Ltd. (Xinxiang, Henan, China). Amylose percentage of PS is 14.3%, and its amylose/amylopectin ratio is 0.17. Relatively crystallinity of PS is 26.2%, and its X-ray diffraction pattern is C type, showing diffraction peaks at 5.7°, 15.2°, 17.2°, and 23.1°. The enzyme (PulY103A) was prepared, and its activity was verified according to the method of Liu et al. [12]. The Megazyme RS assay kit was purchased from Megazyme International (Wicklow, Ireland). Artificial gastric juice (containing pepsin) and artificial small intestinal fluid (containing pancreatin and amyloglucosidase) were provided by Yuanye Co., Ltd. (Shanghai, China).

### 2.2. Preparation and Determination of RS

Five g of native pea starch (PSn) was mixed with 45 mL phosphate buffer (50 mM, pH of 6.5) and maintained at 80 °C for 30 min. The PSn solution was then autoclaved (PSa) or treated with PulY103A prior to autoclaving (PSpa). For PSa preparation, the PSn solution was autoclaved at 121 °C and 15 MPa for 20 min, after which it was cooled and kept at 4 °C overnight. For the combination method (PSpa), the PSn cooled solution was treated with PulY103A (120 units per dry starch basis) at 40 °C for 16 h and stirred at 200 rpm before process of autoclaving. The treated samples were washed with ethanol (two volumes), centrifuged (3000× *g*; 10 min), and washed twice with ethanol. The samples were dried overnight at 50 °C and passed through a 100 mesh sieve. The RS content was determined using the Megazyme RS Assay Kit.

### 2.3. Determination of Amylose Content

The method of Khan et al. was employed to determine the amylose content [13].

### 2.4. Swelling Power and Solubility

The method of Feng et al. was used to investigate the solubility (S) and swelling power (SP) of the samples [14]. Briefly, 0.1 g samples were kept in 5 mL distilled water at 90 °C for 30 min and stirred at 200 rpm. The solutions were then cooled to room temperature and centrifuged (1500× *g*; 10 min). The supernatants were then collected and dried, and the sediments were immediately weighed.

### 2.5. SEM Analysis

The method of Liu et al. was employed for the SEM analysis of starch granules [3]. In brief, samples were treated via freeze drying and gold coating. The SEM analysis was performed at an acceleration voltage of 20 kV.

### 2.6. Thermal Analysis

The thermal analysis of the starch was checked by DSC (Netzsch 200PC, Bavaria, Germany). The procedure was as follows: the temperature was increased from 25 °C to 150 °C at a rate of 10 °C/min and then decreased to 25 °C at rate of 5 °C/min. Samples (5.0 ± 0.1 mg) were weighed in aluminum pans, mixed with 20 μL of deionized water, and sealed. An empty aluminum pan was used as a control.

### 2.7. Fourier Transform Infrared Spectra

The method of Sun et al. was employed for Fourier transform infrared spectra (FTIR) analysis [15]. In brief, KBr (150 mg) was added to 2 mg of each starch sample, and each sample was mixed and pressed to form thin slices. FTIR spectrometry was determined using Nicolet5700 spectrophotometer (Thermo Fisher Nicolet 5700, Waltham, MA, USA), scanning the whole spectrum (4000–500 cm^−1^). Three repetitions were carried out for each sample.

### 2.8. X-ray Diffraction

The method of Wang et al. was employed for X-ray diffraction (XRD) analysis [16]. In brief, X-ray diffractometer (D8Advance, Bruker, Germany) was employed and operated under the following conditions: Cu Ka radiation (wavelength, 0.154 nm); 40 kV, 40 mA; and 5° to 45° (2*θ*) at rate of 5°/min. Three repetitions were carried out for each sample. The relative crystallinity was analyzed by the ratio of the crystalline area to the total diffractogram area.

### 2.9. Preparation of the Bread

The basic dough formula (control) was as follows: wheat flour (200 g), yeast (2.5 g), sugar (25 g), salt (3 g), milk powder (16 g), butter (16 g), and water (136 g). In the experimental bread, the wheat flour was replaced by 5%, 10%, and 15% PSpa (29.8% RS). All the ingredients were mixed in a mixing bowl for about 10–15 min until the dough became soft and elastic. After mixing, the dough was fermented at 28 °C for 1 h, and the fermented doughs were divided into two pieces, molded, placed in pans at 37 °C for 50 min, and then baked at 230 °C for 25 min. In addition, the baked breads were cooled before further testing.

### 2.10. Properties of the Bread

#### 2.10.1. Sensory Analysis

The bread samples were analyzed according to the method of Li et al. with some modifications [17]. The sensory evaluation was performed by 25 trained panelists who were laboratory staff members (from the Faculty of Food Science and Engineering, Kunming University of Science and Technology). The maximum scores for each parameter were as follows: texture: 15, color of crust: 20, appearance: 15, aroma: 25, and taste/flavor: 25.

#### 2.10.2. Color of Crust

The color parameters were determined according to method of Liu et al. [18]. In brief, the color parameters (L, lightness; a, redness; and b, yellowness) of the crust of each bread were measured by a fully automatic colorimeter (WSC-S, Shanghai Inesa Instrument Co., Ltd., Shanghai, China). The samples were taken from the center of the crust of bread. Each sample was taken at random, and three repetitions were carried out.

#### 2.10.3. Hardness

The crumb hardness of the samples was measured according to the method of Liu et al. [18]. The hardness of the samples was evaluated by TA-XT Plus texture analyzer (Lotun Science, Shanghai, China). The samples (2.0 × 2.0 × 2.0 cm) were compressed twice at 1.0 mm/s and 5.0 mm/s before and after the test. The 50% compression level was used by the P/36 probe, and the induction force was 8 g. The hardness was checked five times. The crumb hardness of control samples and breads supplemented with 10% PSpa was detected at time intervals during storage time (0, 1, 3, and 5 days).

#### 2.10.4. Specific Volume

The specific volume was determined according to method of Liu et al. [18]. In brief, the specific volume of bread was measured after the bread loaves were out of the oven for 2 h. The bread volume was determined by the rapeseed displacement method, and the specific volume (mL/g) was calculated according to the formula: *P* = V/m: *P*, where V is the volume (mL) and m is the weight (g). Three repetitions were carried out.

#### 2.10.5. Total Dietary Fiber Content

The total dietary fiber (TDF) content of the bread sample was determined using a TDF determination kit (Megazyme Int.). The samples were dried and ground to determine the TDF content. The results were expressed as the percentage of total dietary fiber on a dry basis (%, g TDF/100 g dry sample).

#### 2.10.6. In Vitro Bread Digestibility

The in vitro digestibility of the bread sample was determined according to the method of Zhang et al. with some modifications [19]. Briefly, the ground sample (3.0 g) was treated with 10 mL artificial gastric juice and 27 mL ultrapure water. The mixture was then incubated in a water bath at 37 °C for 1 h, and the solution was adjusted to pH of 6.8 with NaHCO_3_ solution. Then 15 mL artificial small intestinal fluid and 20 mg α-amylase (8 U/mg) were added to the mixture and incubated in water bath at 37 °C for 3 h. The supernatants were taken 0, 30, 60, 90, 120, and 180 min after incubation, and the reduction in sugar content was determined by the 3,5-dinitrosalicylic acid (DNS) method [20]. The hydrolysis index (HI) is the ratio of the area under the hydrolysis curve of the sample to the area under the hydrolysis curve of white bread as a reference sample. The HI and the in vitro glycemic index (GI) were calculated as follows [21]:HI = Area under the curve of the sample/Area under the curve of white bread
GI = 39.71 + 0.549HI

#### 2.10.7. In Vitro Mineral Bioavailability

To determine the in vitro bioavailability of calcium, iron, and zinc, Ca, Fe, and Zn mixtures (500 mg, 100 mg, 6.7 mg) were added to the bread formulation corresponding to the daily intake values stated by the Chinese Society of Nutrition.

In vitro mineral bioavailability is expressed as a ratio of the amount of the mineral released during enzymatic digestion to the total amount of the mineral contained in the sample. For the determination of mineral contained in the sample, the samples were treated with the method of in vitro digestibility above. Then the supernatant (10 mL) after digestion was washed with HNO_3_ (2 mL) in Teflon vessels heated in a microwave oven. Then, it was filtered into a volumetric flask (100 mL). Lanthanum chloride solution (1 mL, 0.1%, *w*/*v*) was added to determine the calcium content. The concentration of minerals was checked by atomic absorption spectrophotometry (AAS; Thermo Scientific (Waltham, MA, USA)).

### 2.11. Statistical Analysis

Statistical analysis was conducted using Origin version 9.0. The experiments were performed in triplicate, and all data are expressed as mean ± standard deviations. One-way analysis of variance (ANOVA) and *t*-test (comparing of two samples) were employed for paired comparison analysis, and Pearson’s regression was employed for a correlation analysis. Comparison of means was performed using Tukey–Kramer HSD test at *p* < 0.05.

## 3. Results and Discussion

### 3.1. RS Preparation

In this study, the RS was prepared by a combination of autoclaving and moderate amylopullulanase hydrolysis. Up until now, there have been few studies of moderate amylopullulanase being applied for RS production [2]. In this study, the RS content increased from 8.2% (PSn) to 29.8% (PSpa) (Appendix A), and the growth rate of the pea RS preparation reached 3.63, which is similar to that in the report of Liu et al. [4] and much higher than that in other reports using pullulanase [22,23], which showed PulY103A has great potential for use in RS production.

### 3.2. RS Properties

The properties of the starch samples are shown in Table 1. The results showed that the amylose content value significantly increased from 14.3% (PSn) to 30.5% (PSpa) due to the debranching ability of pullulanase (*p* < 0.05) [3]. Also, the SP and S showed a decreasing trend and changed significantly (refer to Table 1) (*p* < 0.05). The decreasing trend of SP and S was mainly due to the higher amylose content and molecule rearrangement, which would lead them to form greater crystalline regions and leach less during swelling [2]. The results showed that PSpa (*T*_o_, 96.1 °C; Δ*H*, 10.5 J/g) had a better thermal stability compared to PSa (*T*_o_, 86.8 °C; Δ*H*, 4.5 J/g) and PSn (*T*_o_, 74.1 °C; Δ*H*, 2.5 J/g), which might be attributed to PSpa having more of a double-helix structure and the integrated crystal structure formed during retrogradation and debranching [15,22]. The SEM results of the starch samples showed that PSpa had more angular and flakey shapes compared to those of PSa (Appendix A), indicating that PSpa was much more resistant to enzymatic hydrolysis [3].

### 3.3. FTIR and XRD Analysis

The FTIR spectra are shown in Figure 1a. There were vibrations for the -OH groups (3403, 3411, and 3412 cm^−1^) in all of the samples. And the treated groups had higher wave numbers compared to those of the control, which shows that the control sample has fewer -OH groups, while the treated groups may have more hydrogen bonds in their molecular chains [3,24]. As for the vibrations for the C-H groups (2931, 2927, and 2932 cm^−1^), the peak intensity of the treated groups was lower than that of the control group, which was attributed to molecular disruption [25]. In addition, there was no significant difference between the samples, which showed that no chemical groups were formed under the treatment.

The XRD result is shown in Figure 1b. The crystalline structure of the native pea starch has a C-type pattern (5.7°, 15.2°, 17.2°, and 23.1°), while the PSpa has a B + V complex pattern (5.4°, 17.0°, 19.3°, 22.1°, and 24.0°). There have been a few reports that have shown that the starch crystalline structure changes during processing [3,26]. Moreover, a few reports have found that debranched pea starch has diffractive peaks around 19.5°, which are typical of the V pattern corresponding to the amylose–lipid complex. As a result, PSpa was found to belong to RS types III + V rather than the typical RS type III [3,27,28].

Our results showed that the crystallinity of PSpa reached 26.6%, which was higher than the crystallinity of PSa (20.5%) and PSn (13.3%) (Table 1). The increase in linear chains was probably the main reason for the PSpa having a relatively high crystallinity, which was consistent with its thermal properties.

### 3.4. Bread Properties

Due to the structure and properties of PSpa, PSpa has been applied for white breadmaking in this study. RS types III + V have not been applied in breadmaking yet. The results of our sensory analysis (Table 2) showed that there was no significant difference among the samples when the content of the PSpa in the bread formula did not go over 10% (*p* > 0.05). However, large differences appeared in some parameters, including its appearance, mouthfeel, texture, and overall acceptability when the content of the PSpa reached 15% (*p* < 0.05). Moreover, the bread qualities are shown in Table 3, which shows that the properties of the breads with PSpa were not significantly influenced compared with those of the control group when the content of PSpa reached 10% (*p* > 0.05). Also, the parameter of crust color did not change too much when the concentration of PSpa was below 15% (*p* > 0.05). The specific volume decreased significantly when the content of PSpa reached 15%. Meanwhile, the hardness value increased by about 70% in comparison with the control at the same concentration. Thus, the results of the breads’ qualities were consistent with the sensory analysis (*p* < 0.05), which indicated that the low concentration of PSpa (<15%) added for breadmaking did not affect the bread quality very much. This phenomenon is similar to that in other reports [9,29,30]. The decrease in the bread specific volume at PSpa levels above 10% was probably attributed to the fact that the yeast is unable to utilize RS during fermentation, which would slow down the fermentation process. In addition, the undamaged starch would cause the gas cells to distribute unevenly. As for the result, the hardness of the bread was negatively correlated with its specific volume (*p* < 0.05), so a large increase in hardness occurred at the same concentration.

RS is considered as an insoluble fiber that is resistant to enzyme hydrolysis due to its compact structure. Among the five types of RS, RS type V is much more heat-stable and difficult to digest due to lipids binding to amylose in granules [31,32]. Our results showed that the PSpa-supplemented bread has a significantly increased TDF content compared to the control (Table 4). It is possible that the high fiber content and properties of PSpa are physical or chemical barriers to the enzyme; thus, the GI value decreased significantly in the PSpa-supplemented bread compared to the control (*p* < 0.05). As shown in Table 4, the bread sample changed from a high-GI food (GI ≥ 70) to a medium-GI one (GI 56–69) in the 10% PSpa-supplemented bread at day 0. This result was similar to that in the report of Aribas et al. and indicates that consuming bread with PSpa could cause lower levels of digestibility and insulin response [21]. However, the RS-supplemented breads reported were still a high-GI food at day 0 [21,33,34]. Also, Table 4 shows the result of the in vitro bioavailability of theminerals of the bread samples. The Ca, Fe, and Zn bioaccessibility values of the white bread were 40.2%, 18.9%, and 24.8%, respectively. Moreover, the bread sample supplemented with 10% PSpa caused a small but significant decrease in Ca bioavailability (*p* < 0.05), and there were not large differences in the Fe and Zn bioavailability between the bread sample supplemented with 10% PSpa and the control (*p* > 0.05). Generally, dietary fiber can cause a decrease in the bioavailability of minerals; however, the addition of PSpa only lightly influenced the bioavailability of the minerals in the bread samples, which is similar to a report of RS type IV supplemented breads that caused small but significant increases in Ca and Zn bioavailability, and no significant difference of Fe bioavailability compared to control samples [21].

The crumb hardness is correlated with starch retrogradation [35]. Thus, the crumb hardness of the control samples and breads supplemented with 10% PSpa was checked when they were being stored. As shown in Appendix A, an increasing trend was found in all of the samples, and there were no significant differences between the control samples and breads supplemented with 10% PSpa (*p* > 0.05) at the beginning and on the first day of storage. These results might contribute to PSpa having the structure of starch–lipid complexes, which inhibits the rate of starch retrogradation [35]. Generally, the hardness of the RS-supplemented breads obviously increased and became very different from the control during storage [21,33].

## 4. Conclusions

A moderate amylopullulanase (PulY103A) was shown to be efficient in pea RS preparation. The prepared pea RS (PSpa) belongs to RS types III + V and has good properties. Furthermore, we applied PSpa in breadmaking, and our results showed that the properties of the breads supplemented with PSpa were not significantly influenced when the content of RS was under 10%. Moreover, the bread sample changed to a medium-GI food with 10% PSpa supplementation and changed slightly in terms of its mineral bioavailability at the same concentration compared to that of other dietary fibers, which indicates that PSpa has great potential for use in dietary foods. Moreover, other kinds of dietary foods should be explored in order to promote the wide application of PSpa.

## Figures and Tables

**Figure 1 foods-13-01251-f001:**
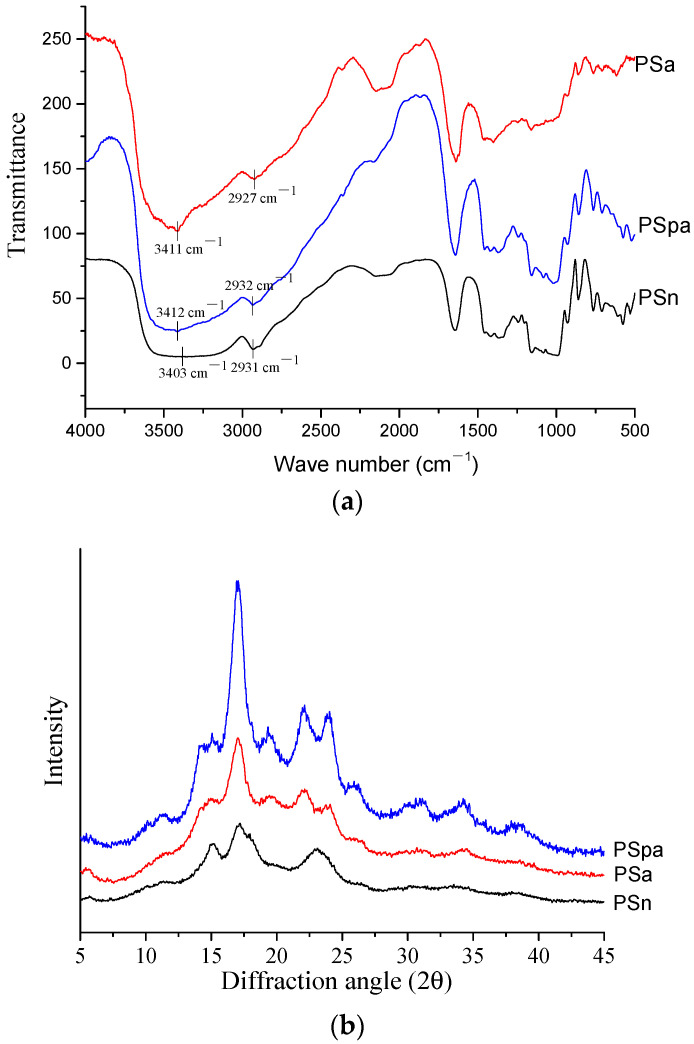
FTIR spectra (**a**) and XRD patterns (**b**) of native and treated pea starches. PSn, Native pea starch; PSa, Pea starch treated with autoclaving; and PSpa, Pea starch treated with debranching and autoclaving.

**Table 1 foods-13-01251-t001:** The properties of pea starch samples.

Parameters	PSn	PSa	PSpa
Amylose content (%)	14.3 ± 0.2 ^c^	20.7 ± 0.1 ^b^	30.5 ± 0.1 ^a^
Solubility (%)	43.5 ± 1.4 ^a^	31.0 ± 1.2 ^b^	12.5 ± 0.1 ^c^
Swelling power (%)	21.5 ± 2.7 ^a^	11.0 ± 0.7 ^b^	7.9 ± 0.8 ^c^
Relatively crystallinity (%)	13.3	20.5	26.6
*T*_o_ (°C)	74.1	86.8	96.1
*T*_p_ (°C)	99.6	113.7	121.3
*T*_c_ (°C)	110.0	120.6	133.3
Δ*H* (J/g)	2.5	4.5	10.5

^a–c^, Different letters within the same index are significantly different (*p* < 0.05). PSn, Native pea starch; PSa, Pea starch treated with autoclaving; and PSpa, Pea starch treated with debranching and autoclaving.

**Table 2 foods-13-01251-t002:** Sensory evaluation of bread samples.

Samples	Appearance	Aroma	Taste	Texture	Colour
Control	13.3 ±1.2 ^a^	21.8 ± 1.9 ^a^	20.7 ± 1.4 ^a^	13.4 ± 1.0 ^a^	17.4 ± 0.8 ^a^
5% PSpa	13.1 ± 0.9 ^a^	22.1 ± 1.3 ^a^	21.1 ± 1.3 ^a^	13.0 ± 0.9 ^a^	17.0 ± 1.0 ^a^
10% PSpa	12.9 ± 1.4 ^a^	21.0 ± 3.0 ^a^	20.8 ± 1.9 ^a^	13.1 ± 1.2 ^a^	16.9 ± 1.0 ^a^
15% PSpa	10.4 ± 1.6 ^b^	18.8 ± 3.5 ^b^	14.5 ± 1.7 ^b^	10.6 ± 1.2 ^b^	16.6 ± 1.5 ^b^

All values are the mean ± standard deviation (*n* = 25). ^a,b^, Different letters within the same index are significantly different (*p* < 0.05).

**Table 3 foods-13-01251-t003:** The properties of bread samples.

Samples	Specific Volume (mL/g)	Color of Crust			Hardness (g)
		L*	a*	b*	
Control	3.43 ± 0.02 ^a^	12.26 ± 0.04 ^a^	−0.09 ± 0.01 ^b^	−0.64 ± 0.01 ^a^	234.1 ± 14.8 ^b^
5% PSpa	3.47 ± 0.03 ^a^	12.21 ± 0.09 ^a^	−0.10 ± 0.01 ^b^	−0.67 ± 0.04 ^a^	240.9 ± 31.7 ^b^
10% PSpa	3.48 ± 0.01 ^a^	12.17 ± 0.04 ^a^	−0.11 ± 0.03 ^b^	−0.67 ± 0.02 ^a^	261.7 ± 24.4 ^b^
15% PSpa	2.78 ± 0.03 ^b^	12.19 ± 0.10 ^a^	−0.06 ± 0.01 ^a^	−0.73 ± 0.02 ^b^	439.0 ± 39.0 ^a^

^a,b^, Different letters within the same index are significantly different (*p* < 0.05).

**Table 4 foods-13-01251-t004:** The TDF, in vitro mineral bioavailability, and in vitro GI of bread samples.

Samples	TDF (%)	In Vitro Mineral Bioavailability (%)			In Vitro GI
		Ca	Fe	Zn	
Control	3.1 ± 0.1 ^b^	40.3 ± 0.7 ^a^	18.9 ± 2.4 ^a^	23.6 ± 0.1 ^a^	100.0 ± 0.0 ^a^
10% PSpa	11.6 ± 0.1 ^a^	32.7 ± 0.8 ^b^	17.3 ± 1.5 ^a^	22.6 ± 0.6 ^a^	69.9 ± 0.5 ^b^

All values are the mean ± standard deviation (*n* = 3). ^a,b^, Different letters within the same index are significantly different (*p* < 0.05).

## Data Availability

The original contributions presented in the study are included in the article/Appendix A, further inquiries can be directed to the corresponding author.

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
