# Peer review of "Preparation of Resistant Starch Types III + V with Moderate Amylopullulanase and Its Effects on Bread Properties"

_foods, 2024, doi:10.3390/foods13081251_

Round 1
Reviewer 1 Report
Comments and Suggestions for Authors
Comments and Suggestions for Authors,
The manuscript “Resistant starch â…¤ preparation with moderate amylopullulanase and its effects on bread properties”, is generally well written; however, it must be corrected in order to improve its writing and quality based on the following specific comments:
Specific comments:
Lines 65-68, The authors should indicate more information about the physicochemical characteristics of pea starch. For example: Amylose percentage, amylose/amylopectin ratio, percentage (crystallinity index), type of X-ray diffraction pattern, etc.
Line 71, Should read as: Five g of native pea starch (PSn) …
Line 72, Should read as: … maintaining at 80 °C for 30 min.
Line 74, What was the pressure generated by the autoclave process at 121 °C? Please clarify!
Lines 77 and 78, should read as: centrifuged as follows (3,000 ´ g; 10 min) and …
Line 87, Should read as: … and centrifuged (1,500 ´ g; 10 min).
Line 94, Should read as: Samples (5.0 ± 0.1 mg) were …
Line 96, The authors should specify and give more information on the FTIR technique used because it is considered important to provide information on the structural modifications and the characterization of functional groups introduced in the reactants and products obtained, for example: How many samples were analyzed in total? It is not clear whether the spectral analysis was carried out on different samples and the spectra reported are representative of a single sample or are the average spectra of several acquisitions made on different points of the same sample.
Line 99, The authors should indicate more information about this reported technique. Similar to the comment regarding the FTIR technique.
Lines 117-120, I consider it important to provide more information about this determination. For example: How many determinations were made for each sample? How were these determinations evaluated (at random points, at the ends of the cortex, etc.)? The results indicated were considered as the averages? How many repetitions?
Line 150, Should read as: In vitro mineral bioavailability
Line 156, Should read as: … in the sample, the samples were …
Line 158, Should read as: … with HNO3 (2 mL) in Teflon vessels by …
Line 159, Should read as: … a volumetric flask (100 mL).
Lines 162-164, The authors should indicate more information. For example, what type of statistical design did they use (completely randomized design, randomized block design), etc.? What was the number of replicates or repetitions in the analyses? What was the significance level value used (a) to determine significant differences?
Lines 175-177, What was the reason for the amylose content to increase in the PSn and PSPa samples? Please clarify with arguments in accordance with the scientific literature reported on the matter!
Line 308, References Section, please correct the duplicity of the numbers corresponding to each reference cited.
Author Response
Dear Ms. Yvette Yan,
Thank reviewer for the valuable comments. Authors made highlights in the manuscript (red for the reviewers which they want to change). All general and minor errors were fixed following by reviewer’s comments. Especially, we try to make clear corrections. Please check out the list of changes in manuscript.
I look forward to your good news.
Yours sincerely,
Peng Han
Reviewer #1
Q1. Lines 65-68, The authors should indicate more information about the physicochemical characteristics of pea starch. For example: Amylose percentage, amylose/amylopectin ratio, percentage (crystallinity index), type of X-ray diffraction pattern, etc.
A1. Thank you very much for a valuable comment. According to your comment, more detail information has been provided.
Line 69–72; Amylose percentage of PS is 14.3%, and its amylose/amylopectin ratio is 0.17, Relatively crystallinity of PS is 26.2%, and its X-ray diffraction pattern is C-type showing diffraction peaks at 5.7°, 15.2°, 17.2°, and 23.1°.
Q2. Line 71, Should read as: Five g of native pea starch (PSn) …
Line 72, Should read as: … maintaining at 80 °C for 30 min.
Lines 77 and 78, should read as: centrifuged as follows (3,000 ´ g; 10 min) and …
Line 87, Should read as: … and centrifuged (1,500 ´ g; 10 min).
Line 94, Should read as: Samples (5.0 ± 0.1 mg) were …
Line 150, Should read as: In vitro mineral bioavailability
Line 156, Should read as: … in the sample, the samples were …
Line 158, Should read as: … with HNO3 (2 mL) in Teflon vessels by …
Line 159, Should read as: … a volumetric flask (100 mL).
A2. Thank you very much for a valuable comment. According to your comment, the changes have been made.
Q3. Line 74, What was the pressure generated by the autoclave process at 121 °C? Please clarify!
A3. Thank you very much for a valuable comment. According to your comment, the information has been provided.
Q4. Line 96, The authors should specify and give more information on the FTIR technique used because it is considered important to provide information on the structural modifications and the characterization of functional groups introduced in the reactants and products obtained, for example: How many samples were analyzed in total? It is not clear whether the spectral analysis was carried out on different samples and the spectra reported are representative of a single sample or are the average spectra of several acquisitions made on different points of the same sample.
A4. Thank you very much for a valuable comment. According to your comment, more detail information has been provided.
Line 109–112; In brief, KBr (150 mg) was added to 2 mg of starch sample, and the sample was mixed and pressed to form thin slices. FTIR spectrometer was performed by Nicolet5700 spectrophotometer (Thermo Fisher Nicolet 5700, USA) scaning the whole spectrum (4000–500 cm−1). Three repetitions were carried out for each sample.
Q5. Line 99, The authors should indicate more information about this reported technique. Similar to the comment regarding the FTIR technique.
A5. Thank you very much for a valuable comment. According to your comment, more detail information has been provided.
Line 115–119; In brief, X-ray diffractometer (D8Advance, Bruker, Germany) was employed, and operated as follow conditions: Cu Ka radiation (wavelength, 0.154 nm); 40 kV, 40 mA; 5° to 45° (2θ) at rate of 5°/min. Three repetitions were carried out for each sample.
Q6. Lines 117-120, I consider it important to provide more information about this determination. For example: How many determinations were made for each sample? How were these determinations evaluated (at random points, at the ends of the cortex, etc.)? The results indicated were considered as the averages? How many repetitions?
A6. Thank you very much for a valuable comment. According to your comment, more detail information has been provided.
Line 140–141; Each sample was taken at random, and three repetitions were carried out.
Q7. Lines 162-164, The authors should indicate more information. For example, what type of statistical design did they use (completely randomized design, randomized block design), etc.? What was the number of replicates or repetitions in the analyses? What was the significance level value used (a) to determine significant differences?
A7. Thank you very much for a valuable comment. According to your comment, more detail information has been provided.
Lines 187-188, The experiments were performed in triplicate and all data were expressed as mean ± standard deviations.
Lines 191-192, Comparison of means was performed by Tukey-Kramer HSD at P < 0.05.
Q8. Lines 175-177, What was the reason for the amylose content to increase in the PSn and PSpa samples? Please clarify with arguments in accordance with the scientific literature reported on the matter!
A8. Thank you very much for a valuable comment. According to your comment, more detail information has been provided.
Lines 205-206, amylose content value had a significant increasing from 14.3% (PSn) to 30.5% (PSpa) due to the debranching ability of pullulanase (P < 0.05) [7].
Q9. Line 308, References Section, please correct the duplicity of the numbers corresponding to each reference cited.
A9. The mistakes were corrected.

Reviewer 2 Report
Comments and Suggestions for Authors
Enzymatically modified starch by shortening the chains should be ranked as RS3 (not RS5, amylose-lipid complexes, it not obvious from the given data) because RS3 supports retrogradation. The introduction is all confused, it does not describe the essence of the issue. The references for the methods are not given correctly. Considering that the starch comes from legumes, it would be advisable to measure the content of rapidly digestible, slowly digestible and resistant starch in the monitored material. Or at least the fiber content (TDF). As to sensory evaluation of bread – storability is one of the important properties. It is necessary to complete how the sensory properties will be changed after e.g. 3 days. It is important esp. because RS3 changes retrogradation properties. In details see *.pdf file.

Author Response
Dear Ms. Yvette Yan,
Thank reviewer for the valuable comments. Authors made highlights in the manuscript (red for the reviewers which they want to change). All general and minor errors were fixed following by reviewer’s comments. Especially, we try to make clear corrections. Please check out the list of changes in manuscript.
I look forward to your good news.
Yours sincerely,
Peng Han
Reviewer #2
Q1. The authors, for some mysterious reason, use Roman numerals to refer to resistant starch. Return please to Arabic numbers. Additionally, the enzymatically modified starch by shortening the chains is ranked as RS3 (not RS5, amylose-lipid complexes, it not obvious from the given data) because RS3 supports retrogradation
A1. Thank you very much for a valuable comment. However, the peak at 19.2° is a typical one of V pattern corresponding to the amylose-lipid complex. This result indicated that PSpa belongs to RS3+5 rather than a typical RS3 (Liu et al., 2023). Furthermore, some changes have been made acoording to the RS5.
Liu Z.; Liu, L.; Han, P.; Liang, X. Pea resistant starch preparation with cold-active type I pullulanase from Bacillus megaterium and its potential application in rice noodles. LWT-Food Sci. Technol. 2023, 182, 114799. https://doi.org/10.1016/j.lwt.2023.114799.
Q2. The pea resistant starch (PSpa) produced with PulY103A had … better thermostability – no evidence for this.
A2. This sentence has been changed; … better thermostability than native pea starch (PSn) and PS autoclaved (PSa).
Q3. The introduction is all confused, it does not describe the essence of the issue.
A3. Thank you very much for a valuable comment. According to your comment, the introduction has been modified.
Q4. Line 40: special.
A4. The mistake has been corrected.
Q5. Line 41: Five kinds of RS have been applied in dietary foods [7–9]. – it was written in the previous paragraph. Neither delete this sentence or move it in front of “RS is classified into five groups,…”
A5. Thank you very much for a valuable comment. According to your comment, the sentence was put it in front of “RS is classified into five groups,…”.
Q6. Line 42: Among these foods, bread is consumed… - among what foods?
A6. The sentence has been modified. Among these dietary foods…
Q7. Lines 43-45; the bread is usually made from refined wheat flour by mixing with gluten, other cereals, and different types of RS – usually mixing with gluten and RS?
A7. The sentence has been modified.
Lines 46-47; the bread has been made from refined wheat flour by mixing with gluten, other cereals, and different types of RS.
Q8. Line 50: Legumes are one of the main crops for starch production. – it is not correct. The production of starch from pea and other legumes in the world is not high.
A8. The sentence has been modified.
Line 53; Legumes are one of resources for starch production.
Q9. What amount of lipids were found in pea starch? The references for the methods are not given correctly.
A9. The label of pea starch shows that there are no lipids in the stuff, and the amount of lipids of pea starch was not checked by our group. However, the result of XRD showed that PSpa has amylose-lipid structure. Of course, it is a good point to concern.
The references for the methods have been checked and corrected.
Q10. 2.2. Preparation and determination of RS; Line 73: For autoclaving, the PS solution…; Line 77: The treated samples were dealed with ethanol…???
A10. The sentence “For autoclaving, the PS solution…” has been changed as “For Psa preparation, The PSn solution…”.
Some reports showed that the ethanol was used to precipitate the RS sample (Thakur et al., 2021; Li et al., 2019).
Thakur, M., Sharma, N., Rai, A. K., & Singh, S. P. (2021). A novel cold-active type I pullulanase from a hot-spring metagenome for effective debranching and production of resistant starch. Bioresource Technology, 320, 124288.
Li, L., Yuan, T. Z., Setia, R., Raja, R. B., Zhang, B., & Ai, Y. (2019). Characteristics of pea, lentil and faba bean starches isolated from air-classified flours in comparison with commercial starches. Food Chemistry, 276, 599-607.
Q11. 2.3. Determination of amylose content
Line 82: The amylose content of starch was determined by the method of Liua et al. [4]. – but in Liua et al. you read: The amylose content of starch was determined according to the method of Khan et al. (2020).
A11. The mistake has been corrected.
Line 90; The amylose content of starch was determined by the method of Khan et al. [14].
- Khan, A.; Siddiqui, S.; Ur Rahman, U.; Ali, H.; Saba, M.; Azhar, F.A.; Rehman, M.M.U.; Shah, A.A.; Badshah, M.; Hasan, F.; Khan, S. Physicochemical properties of enzymatically prepared resistant starch from maize flour and its use in cookies formulation. Int. J. Food Prop. 2020, 23, 549–569. https://doi.org/10.1080/10942912.2020.1742736.
Q12. 2.4. Swelling power and solubility
Line 87: Thesupernatants
A12. The mistake has been corrected.
Q13. 2.5. Morphological property
Line 90: The morphology of starch granules was determined by the method of Liua et al. [4]. – it is no Liua method, just SEM – where can we look at the microphotographs?
A13. The mistake has been corrected.
Q14. 2.8. X-ray diffraction
complete the calculation of the relative crystallinity
A14. Thank you very much for a valuable comment. According to your comment, the information has been provided.
Lines 117-119; The relative crystallinity was analyzed by the ratio of the crystalline area to the total diffractogram area.
Q15. 2.9. Preparation of the bread
Line 104: …15% amount of PSpa (30% RS) – really PSPa (see Line 73) or dried product?
A15. Actually, PSpa was the dried product. The part of 2.2 has been changed.
Q16. 2.10.6. In vitro bread digestibility
Line 139: define the artificial gastric juice
Line 141: define the artificial small intestinal fluid, or it was originated fluid from the previous sentence?
Line 147: define what is GI
A16. Thank you very much for a valuable comment. According to your comment, more detail information has been provided.
Lines 74-76; Artificial gastric juice (containing pepsin) and artificial small intestinal fluid (containing pancreatin and Amyloglucosidase) were provided by Yuanye Co., Ltd (Shanghai, China).
Q17. 2.10.7. In vitro mineral bioavailability
Lines 156-157: …in the sample, The samples were dealed with the method of in vitro digestibility above. – lowercase t, dealed?
A17. The mistake has been corrected.
Q18. Other important data on the obtained starch are missing in this section. Considering that the starch comes from legumes, it would be advisable to measure the content of rapidly digestible, slowly digestible and resistant starch in the monitored material. Or at least the fiber content (TDF). As to sensory evaluation of bread – storability is one of the important properties. It is necessary to complete how the sensory properties will be changed after e.g. 3 days. It is important esp. because RS3 changes retrogradation properties.
A18. Thank you very much for a valuable comment. However, the fiber content of RS samples has not been checked. Moreover, the RS application effect on bread is the main point of this study, so the TDF of bread samples were checked.
The crumbs hardness is correlated with starch retrogradation, Thus, the crumbs hardness of control samples and breads supplemented with 10% PSpa was checked during storage time.
Lines 146-148; The crumbs hardness of control samples and breads supplemented with 10% PSpa was detected at time intervals during storage time (0, 1, 3, and 5 days).
Lines 329-337; The crumbs hardness is correlated with starch retrogradation [34]. Thus, the crumbs hardness of control samples and breads supplemented with 10% PSpa was checked during storage time. As showed in Fig. S2 that the increasing trend was found in all samples, and there were no significant differences between the control samples and breads supplemented with 10% PSpa (P > 0.05) at the beginning and 1st day of storage time. These results might be contributed to PSpa has the structure of starch-lipid complexes, which inhibite the rate of starch retrogradation [34]. Generally, the hardness of RS supplemented breads increased obviously, and became giant different from the control during the whole storage time [22,32]. Thus, RS III+â…¤ is more suitable for bread making than RS III.
Lines 449-450; 32. Arp, C.G.; Correa, M.J.; Ferrero, C. High-amylose resistant starch as a functional ingredient in breads: a technological and microstructural approach. Food Bioprocess Tech. 2018, 11, 2182-2193. https://doi.org/10.1007/s11947-018-2168-4.
Lines 454-456; 34. Ma, M.; Mu, T.; Sun, H.; Zhou, L. Evaluation of texture, retrogradation enthalpy, water mobility, and anti-staling effects of enzymes and hydrocolloids in potato steamed bread. Food Chem. 2022, 368, 130686. https://doi.org/10.1016/j.foodchem.2021.130686.
Q19. 3.1. RS preparation
lines 168-170: Up to now, there was only one moderate amylopullulanase from Lactobacillus amylophilus GV6 was used applied for RS production and the growth rate of RS preparation reached at 1.79 – up to now?, 2 x was, used x applied, the unit of growth rate?
line 171: PSPa – the same remark as for the line 104, what is growth rate?
A19. The sentence “Up to now, there was few moderate amylopullulanase being applied for RS production” has changed as “Up to now, there was few moderate amylopullulanase being applied for RS production”. And the growth rate of RS was ratio of the RS content of treated sample to the RS content of native starch.
Q20. 3.2. RS properties
line 177: Also, the SP and S showed a decreasing trend, and changed significantly (P < 0.05). where are the data?, refer to Table 1
A20. The sentence has been modified as “Also, the SP and S showed a decreasing trend, and changed significantly refer to Table 1 (P < 0.05).”.
Q21. 3.3. FTIR and XRD analysis
Figure 1 seems to be identical with the figures in:
- Liua Z.; Liu, L.; Han, P.; Liang, X. Pea resistant starch preparation with cold-active type I pullulanase from Bacillus 315 megaterium and its potential application in rice noodles. LWT-Food Sci. Technol. 2023, 182, 114799. 316 https://doi.org/10.1016/j.lwt.2023.114799.
Han is also the co-author of this paper.
FTIR is not described in details. Changes?
lines 224-226: Moreover, few reports found that the debranched pea starch has diffractive peaks around 19.0°, which is a typical one of V pattern corresponding to the amylose-lipid complex [4]. – it is the same reference from your team
A21.The results of FTIR and XRD were similar to the report by our team. Although the pullulanases used were so different, the properties of RS prepared were similar. So the results were not described too much. Of course, there were little differences between the data of two reports such as wave number of the peak in FTIR and intensity of diffractive peaks in XRD.
Q22. 3.4. Bread properties
line 238: There has not been any RS â…¤ being applied in bread making yet – it is not true. Starches treated by pollulanase were used for baking – see Web of science.
lines 264-265: Among five types of RS, RS â…¤ is much more heat stable and difficult to digest due to lipids bind to amylose in the granule [29,30]. – What is there any evidence of amylose-lipid complexes? Have you found it by FTIR?
- Conclusions
line 287: The prepared pea RS (PSpa) belongs to the RS â…¤ and has good properties. – no evidence for this
A22. Some changes have been made according to the comments above. The answer of these questions was same as A1.

Reviewer 3 Report
Comments and Suggestions for Authors
The manuscript objectives could be more precise; there are different approaches using Rs, and this lacks sufficient novelty; despite this, as it is written, it lacks sufficient depth. Overall, the work must clearly show an application or possible use of the information more than molecular and chemical composition differences. This work could benefit from additional structural or molecular-related parameters to improve their impact on the carbohydrate community. The literature analysis needs to be more specific and sufficiently in-depth, as well as comparisons. The manuscript should present and analyze the relevant studies, compare and contrast their findings and methods, and highlight their strengths and limitations. Is there a reason why the authors did not disclose the whole data set? It would be interesting to observe the trend when using lower substitution values since it does not make much sense to show only the bread with the highest substitution, which also presented the poorest performance in bread volume (a definitive factor in bread). I recommend improving the statistical analysis; for example, with that amount of data, a robust statistical analysis that integrates their composition or other characteristics could provide a more profound explanation of the effect of treatments in the analyzed material. A PCA analysis or some multivariate analysis could be useful.
Author Response
Dear Ms. Yvette Yan,
Thank reviewer for the valuable comments. Authors made highlights in the manuscript (red for the reviewers which they want to change). All general and minor errors were fixed following by reviewer’s comments. Especially, we try to make clear corrections. Please check out the list of changes in manuscript.
I look forward to your good news.
Yours sincerely,
Peng Han
Reviewer #3
Q1. The manuscript objectives could be more precise; there are different approaches using Rs, and this lacks sufficient novelty; despite this, as it is written, it lacks sufficient depth. Overall, the work must clearly show an application or possible use of the information more than molecular and chemical composition differences. This work could benefit from additional structural or molecular-related parameters to improve their impact on the carbohydrate community. The literature analysis needs to be more specific and sufficiently in-depth, as well as comparisons. The manuscript should present and analyze the relevant studies, compare and contrast their findings and methods, and highlight their strengths and limitations.
A1. Thank you very much for a valuable comment. According to your comment, more information has been provided.
Lines 315-316; However, the RS supplemented breads reported were still high GI food at the 0 day [22,32,33].
Lines 324-325; RS IV supplemented breads that caused small but significant increases in Ca and Zn bioavailability, and no significant difference compared to the control samples [22].
The crumbs hardness is correlated with starch retrogradation, Thus, the crumbs hardness of control samples and breads supplemented with 10% PSpa was checked during storage time.
Lines 146-148; The crumbs hardness of control samples and breads supplemented with 10% PSpa was detected at time intervals during storage time (0, 1, 3, and 5 days).
Lines 329-337; The crumbs hardness is correlated with starch retrogradation [34]. Thus, the crumbs hardness of control samples and breads supplemented with 10% PSpa was checked during storage time. As showed in Fig. S2 that the increasing trend was found in all samples, and there were no significant differences between the control samples and breads supplemented with 10% PSpa (P > 0.05) at the beginning and 1st day of storage time. These results might be contributed to PSpa has the structure of starch-lipid complexes, which inhibite the rate of starch retrogradation [34]. Generally, the hardness of RS supplemented breads increased obviously, and became giant different from the control during the whole storage time [22,32]. Thus, RS III+â…¤ is more suitable for bread making than RS III.
Lines 449-450; 32. Arp, C.G.; Correa, M.J.; Ferrero, C. High-amylose resistant starch as a functional ingredient in breads: a technological and microstructural approach. Food Bioprocess Tech. 2018, 11, 2182-2193. https://doi.org/10.1007/s11947-018-2168-4.
Lines 451-453; 33. Sciarini, L.S.; Bustos, M.C.; Vignola, M.B.; Paesani, C.; Salinas, C.N.; Perez, G.T. A study on fibre addition to gluten free bread: Its effects on bread quality and in vitro digestibility. J. Food Sci. Technol. 2017, 54, 244-252. https://doi.org/10.1007/s13197-016-2456-9.
Lines 454-456; 34. Ma, M.; Mu, T.; Sun, H.; Zhou, L. Evaluation of texture, retrogradation enthalpy, water mobility, and anti-staling effects of enzymes and hydrocolloids in potato steamed bread. Food Chem. 2022, 368, 130686. https://doi.org/10.1016/j.foodchem.2021.130686.
Q2. Is there a reason why the authors did not disclose the whole data set?
A2. If the readers need to see the data, we will disclose it.
Q3. It would be interesting to observe the trend when using lower substitution values since it does not make much sense to show only the bread with the highest substitution, which also presented the poorest performance in bread volume (a definitive factor in bread).
A3. The concentration of substitution was designed according to the literatures. Moreover, the results showed that the specific volume was not changed significantly compared with the control group when the content of PSpa was below 15% (P > 0.05).
Q4. I recommend improving the statistical analysis; for example, with that amount of data, a robust statistical analysis that integrates their composition or other characteristics could provide a more profound explanation of the effect of treatments in the analyzed material. A PCA analysis or some multivariate analysis could be useful.
A4. Thank you very much for a valuable comment. According to your comment, Pearson's regression analysis was performed.

Reviewer 4 Report
Comments and Suggestions for Authors
Dear Editor-in-Chief,
This paper compares the effects of a new resistant starch on bread properties. The topic is interesting and novel. But many basic questions should be answered as follows.
Detailed comments;
Why didn't study the effect of the enzyme without autoclaving effect?
Concerning Duncan's letters, the large numbers should be mentioned in the first letters.
Keywords should be arranged alphabetically.
Which type of bread did you make? The plain bread doesn't contain milk powder as well as butter!
For bread analysis, please indicate references for color, hardness, specific volume, and dietary fiber.
Sensory evaluation of bread should be present as a radar plot.
Why didn't measure crumb bread color?
According to table 1 data, PSn is suitable for bread formulation, whereas you applied PSpa. Could you please explain that?
Regards,
Author Response
Dear Ms. Yvette Yan,
Thank reviewer for the valuable comments. Authors made highlights in the manuscript (red for the reviewers which they want to change). All general and minor errors were fixed following by reviewer’s comments. Especially, we try to make clear corrections. Please check out the list of changes in manuscript.
I look forward to your good news.
Yours sincerely,
Peng Han
Reviewer #4
Q1. Why didn't study the effect of the enzyme without autoclaving effect?
A1. The RS of pea starch prepared by the enzyme only has been done, however, the result was unsatisfactory.
Q2. Concerning Duncan's letters, the large numbers should be mentioned in the first letters. Keywords should be arranged alphabetically.
A2. Thank you very much for a valuable comment. According to your comment, some changes have been made.
Q3. Which type of bread did you make? The plain bread doesn't contain milk powder as well as butter!
A3. Milk and butter bread.
Q4. For bread analysis, please indicate references for color, hardness, specific volume, and dietary fiber.
A4. Thank you very much for a valuable comment. According to your comment, the references have been given.
Q5. Sensory evaluation of bread should be present as a radar plot.
A5. Thank you very much for a valuable comment. However, the correlation between the results of Sensory evaluation and the properties of bread samples. Thus, it is convenient to read in the table way.
Q6. Why didn't measure crumb bread color?
A6. The parameters of crumb color checked were not changed obviously, and the correlation between the results of color of sensory evaluation and the crust color of properties of bread samples were analysed. Thus, the results of crumb color were not showed in the paper.
Q7. According to table 1 data, PSn is suitable for bread formulation, whereas you applied PSpa. Could you please explain that?
A7. The inveatigation of potential of RS in low GI food making is the purpose of this study. Although, PSn is suitable for bread formulation, it is not good for low GI food making.

Round 2
Reviewer 2 Report
Comments and Suggestions for Authors
Some comments were corrected, but most important ones remained uncorrected. The introduction is not coherent, many claims in the article are not supported by facts. The authors apparently tried to respond too quickly to comments, but this is at the expense of quality. Again, I do not recommend the article for publication.
Author Response
Dear Ms. Yvette Yan,
Thank reviewer for the valuable comments. Authors made highlights in the manuscript (red for the reviewers which they want to change). All general and minor errors were fixed following by reviewer’s comments. Especially, we try to make clear corrections. Please check out the list of changes in manuscript.
I look forward to your good news.
Yours sincerely,
Peng Han
Q1. The introduction is not coherent.
A1. Thank you very much for a valuable comment. According to your comment, the introduction was improved.
Lines 28-70; Resistant starch (RS) is a kind of starch that could not be digested by enzymes in small intestine. RS is classified into five groups, including RS I–RS â…¤, i.e. RS I, physically inaccessible starch; RS II, uncooked or partially cooked native starch from green banana and sweet potato; RS III, retrograded or recrystallized starch; RS IV; chemically modified starch; RS â…¤, amylose-lipid complex starch [1]. For environmentally friendly and cost-effective, RS production by enzymatic method was preferred especially with moderate enzyme. Pullulanase was the main point for RS production. However, there were few reports about moderate pullulanases being applied on RS production [2–4].
RS plays an important role as dietary fiber on people's health and texture of functional food, which is different from the traditional ones [5]. Five kinds of RS have been applied in dietary foods [6–8]. Among these dietary foods, bread is consumed in many different forms around the world as a source of starch, minerals, and fiber. For better texture and nutritional properties, the bread has been made from refined wheat flour by mixing with gluten, other cereals, and different types of RS. Nowdays, four types of RS (viz, RS I, RS II, RS III, and RS IV) have been tried to be used in bread baking [7,9]. However, it was found that the sensory and textural properties of bread would be negatively affected by increasing the RS content, such as crumb stiffening and dark color [10]. Also, some research found that several RS at certain levels didn’t significantly affect the sensory quality of the bread [9].
Legumes are one of resources for starch production. Among legume starches, pea starch is a good material for RS production, and pea RS has better properties compared to native starch [3]. However, there is no report on pea RS cooperation in bread making. Meanwhile, there are several reports on pea RS prepared by enzymatic method [11], but there is only one report about pea risistant starch prepared by moderate enzyme (PulY103B, type I pullulanase, from Bacillus megaterium) [3]. The research showed that pea RS produced by PulY103B was efficient with high growth rate, and the RS has special stucture and properties including low swelling power and solubility, and good thermostability [3]. In addition, our research group found a moderate amylopullulanase (PulY103A) originated from Bacillus megaterium with α-1,4-transfer activity. The enzyme has optimal reaction conditions at pH 6.5 and 45 °C, almost 70% of its maximum activity was retained at 35 °C, indicating its potential in RS production under moderate conditions [12]. In this study, the capacity of pea RS production by PulY103A was investigated. In addition, the properties of the RS and its potential applications in bread were also studied.
Q2. The evidence and data supporting the identification of resistant starch types within the text are inadequate and have not comprehensively addressed the concerns raised by the reviewers. It is advisable to strengthen the evidentiary base by providing more substantial evidence.
A2. Thank you very much for a valuable comment. According to your comment, two more references which have the similar results and viewpoints were cited.
Lines 28-70; Moreover, few reports found that the debranched pea starch has diffractive peaks around 19.5°, which is a typical one of V pattern corresponding to the amylose-lipid complex. As a result, PSpa was found to belong to the RS III+â…¤ rather than a typical RS III [3,27,28].
- Kang, X.; Gao, W.; Wang, B.; Yu, B.; Guo, L.; Cui, B.; Abd El-Aty, A.M. Effect of moist and dry-heat treatment processes on the structure, physicochemical properties, and in vitro digestibility of wheat starch-lauric acid complexes. Food Chem. 2021, 351, 129303. https://doi.org/10.1016/j.foodchem.2021.129303.
- Maache-Rezzoug, Z.; Zarguili, I.; Loisel, C.; Queveau, D.; Buleon, A. Structural modifications and thermal transitions of standard maize starch after DIC hydrothermal treatment. Carbohyd. Polym. 2008, 74(4), 802-812. https://doi.org/10.1016/j.carbpol.2008.04.047.

Reviewer 3 Report
Comments and Suggestions for Authors
The manuscript is of adequate quality, and the queries have been addressed sufficiently. In my opinion, the manuscript is ready to be published.
Author Response
Dear Pro,
Thank you for your commendation.
Best regards,
Peng Han